# Nailfold Capillaroscopy Analysis Can Add a New Perspective to Biomarker Research in Antineutrophil Cytoplasmic Antibody-Associated Vasculitis

**DOI:** 10.3390/diagnostics14030254

**Published:** 2024-01-24

**Authors:** Gianluca Screm, Lucrezia Mondini, Paola Confalonieri, Francesco Salton, Liliana Trotta, Mariangela Barbieri, Marco Mari, Nicolò Reccardini, Rossana Della Porta, Metka Kodric, Giulia Bandini, Michael Hughes, Mattia Bellan, Selene Lerda, Marco Confalonieri, Barbara Ruaro

**Affiliations:** 1Pulmonology Unit, Department of Medical Surgical and Healt Sciencies, University of Trieste, Hospital of Cattinara, 34149 Trieste, Italy; screm99.gianluca99@gmail.com (G.S.);; 2Department of Experimental and Clinical Medicine, Division of Internal Medicine, University of Florence, Azienda Ospedaliero–Universitaria Careggi (AOUC), 50134 Florence, Italy; 3Division of Musculoskeletal and Dermatological Sciences, Faculty of Biology, Medicine and Health, The University of Manchester, Salford Royal NHS Foundation Trust, Manchester M6 8HD, UK; 4Department of Translational Medicine, Università del Piemonte Orientale (UPO), 28100 Novara, Italy; 5Center for Autoimmune and Allergic Disease (CAAD), Università del Piemonte Orientale (UPO), 28100 Novara, Italy; 6Azienda Ospedaliero–Universitaria, Maggiore della Carità, 28100 Novara, Italy; 7Graduate School, University of Milan, 20149 Milan, Italy

**Keywords:** vasculitis, nailfold video capillaroscopy, ANCA-associated vasculitis, disease severity, microvascular abnormality, autoimmune disease

## Abstract

Background: Antineutrophil cytoplasmic antibody (ANCA)-associated vasculitis (AAV) includes granulomatosis with polyangiitis (GPA), microscopic polyangiitis (MPA), and eosinophilic granulomatosis with polyangiitis (EGPA), all of which are characterised by inflammation of small–medium-sized vessels. Progressive understanding of these diseases has allowed researchers and clinicians to start discussing nailfold video capillaroscopy (NVC) as a future tool for many applications in daily practice. Today, NVC plays a well-established and validated role in differentiating primary from secondary Raynaud’s phenomenon correlated with scleroderma. Nevertheless, there has not been sufficient attention paid to its real potential in the ANCA-associated vasculitis. In fact, the role of NVC in vasculitis has never been defined and studied in a multicentre and multinational study. In this review, we carried out a literature analysis to identify and synthesise the possible role of capillaroscopy for patients with ANCA-associated vasculitis. Methods: Critical research was performed in the electronic archive (PUBMED, UpToDate, Google Scholar, ResearchGate), supplemented with manual research. We searched in these databases for articles published until November 2023. The following search words were searched in the databases in all possible combinations: capillaroscopy, video capillaroscopy, nailfold-video capillaroscopy, ANCA-associated vasculitis, vasculitis, granulomatosis with polyangiitis, EGPA, and microscopic polyangiitis. Results: The search identified 102 unique search results. After the evaluation, eight articles were selected for further study. The literature reported that capillaroscopy investigations documented non-specific abnormalities in 70–80% of AAV patients. Several patients showed neoangiogenesis, capillary loss, microhaemorrhages, and bushy and enlarged capillaries as the most frequent findings. Furthermore, the difference between active phase and non-active phase in AAV patients was clearly discernible. The non-active phase showed similar rates of capillaroscopy alterations compared to the healthy subjects, but the active phase had higher rates in almost all common abnormalities instead. Conclusions: Microvascular nailfold changes, observed in patients affected by vasculitis, may correlate with the outcome of these patients. However, these non-specific abnormalities may help in the diagnosis of vasculitis. As such, new analysis analyses are necessary to confirm our results.

## 1. Introduction

The first European article on vasculitis was published in 1996 [1]. Since that period, there have been many innovations leading researchers to a better understanding of these diseases.

ANCA-associated vasculitis is a heterogeneous group of rare diseases that present necrotising inflammation of small–medium vessels in conjunction with two major antigens targeted by ANCA for proteinase 3 (PR3) or myeloperoxidase (MPO) [2]. Several studies have demonstrated that each of these diseases have many identical clinical manifestations, have many similar histologic features, and may have similar outcomes. There is, however, substantial heterogeneity among these disorders, especially related to the presentation and prognosis of EGPA [3].

PR3-ANCAs are prevalent in GPA, whereas MPO-ANCAs are most commonly present in almost all MPA patients. In EGPA, ANCA can be found in about 40% of patients, and they are correlated with small-vessel vasculitis findings [4].

MPA is necrotising vasculitis that principally affects small vessels and is mainly exhibited as necrotising glomerulonephritis. GPA is necrotising granulomatous inflammation usually including the upper and lower respiratory area instead. Extravascular granulomatosis is a major factor differentiating GPA from MPA [5]. EGPA is eosinophil-rich, and necrotising granulomatous inflammation often includes the lung.

The initial manifestations of subjects with both GPA and MPA typically include non-specific symptoms as fever, myalgias, arthralgias, and a reduction in weight. These prodromal symptoms may be present without evidence of specific organ involvement. Moreover, the progressive development of lesions may include almost any organ or tissue, presenting with symptoms such as cutaneous involvement (typically purpura), upper and/or lower respiratory tract alterations (possible manifestations range from chough to dyspnoea or haemoptysis), urinary abnormalities, or evidence of neurologic disfunctions (peripheral neuropathy is the most common manifestation) [6]. On the other hand, in EGPA, the initial symptoms are completely different and include allergic rhinitis and asthma, commonly followed by an eosinophilic phase that includes peripheral blood eosinophilia and eosinophilic infiltration of multiple organs—these clinical features are the most suitable way to discriminate EGPA from other AACs—the latter typically preceding vasculitis by several years [7].

The diagnosis of vasculitis may be difficult, and tissue biopsy is not always possible. NVC may represent a valuable additional investigative technique, allowing a direct view of microcirculation [6,7,8,9,10]. However, the NVC changes observed in vasculitis are varied and not adequate to be relied upon clinically [7,8,9,10,11]. In addition, insufficient small, uncontrolled observations have described NVC changes in vasculitis. In particular, these studies have used different techniques and methodologies. In addition, isolated capillary abnormalities are also present in healthy subjects.

## 2. Nailfold Video Capillaroscopy

Recently, there has been increased attention on the utility of the non-invasive microvascular inspection of the nailfold bed for several systemic diseases [12,13,14,15,16,17,18,19,20,21,22,23,24,25]. Precisely, nail bed capillaroscopy is useful for assessing subjects for potential scleroderma and other autoimmune diseases [26]. Capillaroscopy is also proposed for non-autoimmune disorders such as arterial hypertension, diabetes mellitus, and in ophthalmologic disorders (e.g., glaucoma) [27,28]. Furthermore, recent discoveries have documented several phenotypes that described by nailfold video- capillaroscopy (NVC), including microangiopathic patterns in many AAV patients, which exhibited a probable correlation with the activity of the pathologies [16].

The NVC is a non-invasive diagnostic instrument used to assess microcirculation and to visualise capillaries in the nailfold bed [8]. The NVC plays a well-established and validated part in discriminating primary (PRP) from secondary (SRP) Raynaud’s phenomenon (RP), which is correlated with scleroderma. Concerning this last topic, NVC has also proven effective in investigating the transition from PRP to SRP. A prospective study demonstrated that RP with negative NVC has the possibility of transiting to SRP during a mean follow-up of 4.4 years, hence the need to strictly monitor PRP patients showing major non-specific alterations in nailfold capillaries at first NVC, at least once a year when at higher risk of transition to SRP [29]. How to interpret its results is a major barrier that has made NVC an underutilised method in many other standard clinical practices, such as the study and evaluation of AAV. The main problem concerning NVC is the large amount of factors that can influence an individual’s nailfold capillary morphology and density; this is also related to a wide intra- and inter-individual variability in healthy subjects. This problem helps us to understand the relevance of a well-suited study group, using the right exclusion criteria to minimise the potential risk of overlapping significant and irrelevant NVC alterations. Moreover, recently, a few studies have used advances in digital imaging technology to evaluate structural parameters and automate this quantitative process. Validation of the fully automated algorithm Automated Nailfold Capillary Counting System (AUTOCAPI)—software for determining the absolute capillary number over 1 linear/mm in NVC images (DS Medica https://www.dsmedica.info accessed on 11 December 2023)—significantly improved the accuracy, reproducibly, and standardisation of NVC results, regarding the assessment of capillary number [30].

Capillaroscopic examinations in the studies were performed using either a stereomicroscope or a digital video capillaroscope. The latter should be preferred due to its greater ease of use.

Capillaroscopic parameters are also studied via qualitative, semiquantitative, and quantitative investigations. Qualitative evaluation mainly describes the global microvascular array, shape, and distribution of the capillaries. Semiquantitative evaluation includes major NVC changes; among those are the giant capillaries, capillary architecture disorganisation, microhaemorrhages, neoangiogenesis, and capillary loss. Quantitative assessments estimate capillary density, avascular areas, diameter of enlarged capillaries, and the frequency of each alteration. A normal capillaroscopic pattern is characterised by the features shown in Table 1 [8,9,10].

Standard NVC can only evaluate morphological capillary abnormalities, but due to the need for techniques that can measure blood perfusion, it has been developed in recent years. For example, laser speckle contrast analysis (LASCA) is a tool (https://www.perimed.it/content/laser-speckle-contrast-analysis-lasca accessed on 11 December 2023) that quantifies the peripheral blood perfusion (PBP) of an area, and it documents PBP during a long period of recording [31,32]. In SSc, the relationships between morphological and functional damage (i.e., decrease in blood flow) have been observed, and these have numerous inferences for both the early diagnosis and progression of the pathology [33]. This technique aims to assess microvascular damage with the advantage of being non-invasive, reliable, reproducible, and quantifiable [31,32,33,34]. However, further observations are required to examine these devices and verify their usefulness in clinical practice, especially regarding AAV.

ANCA-associated vasculitis may be an ideal situation for the utilisation of nailfold capillaroscopy and new techniques due to its active involvement of small vessels including capillaries (with both morphological and functional damage). Capillaroscopy may play a key part in providing useful data for subjects with vasculitis, chiefly during the active phase of the disease.

## 3. Nailfold Video Capillaroscopy in Vasculitis

### 3.1. Characteristics of the Identified Articles

The literature review allowed us to identify eight articles that specifically explored the role of nailfold capillaroscopy in patients affected by ANCA-associated vasculitis. The articles included in the revision are listed in Table 2.

The preliminary analysis of the articles revealed heterogeneity in the capillaroscopy alterations [13,15,17,20] (Figure 1).

In two articles, the authors focused on the study of a GPA cohort only [13,17]. A predominance of the twisting pattern and bleeding were reported in one paper [13] as the main capillaroscopic alterations, in contrast with a single study that described a high proportion (92%) of avascular zones [17]. Furthermore, six studies documented higher rates of capillaroscopic abnormalities in AAV patients but generally non-specific alterations such as increased tortuosity, microhaemorrhages, enlarged and bizarre capillaries, and architectonic disarrangement. Cutolo et al. [10] defined the latter when shapes, length, and diameter vary in continuous loops, which leads to a large alteration in the normal capillary pattern [12,13,14,16,17,18]. Two articles did not report any substantial differences in NVC scores between healthy subjects and AAV patients [15,19].

### 3.2. Capillaroscopic Phenotypes

Capillaroscopy studies have underlined microvascular abnormalities [21,22,23,24,25,26,27,28,29,30,31,32,33,34,35,36,37]. Rarely, strong pathological connotations termed major abnormalities are identified, and these can result in easier and earlier medical diagnosis [10]. Other unusual findings are commonly spread among the population and may include an overlap between healthy controls and people with non-specific diseases, which are called minor abnormalities and have an uncertain pathological meaning; these principally exhibit tortuosity and abnormal shapes.

In detail, there have been several NVC changes detected in AAV patients and reported in scientific articles, which can be resumed as follows:

Capillary density: Evaluation of the number of capillaries per mm of distal row, normal capillary density varies from 9 to 14 capillaries per mm in adults and at least 6 in children. There were no significant density modifications detected in the activity phase and/or remission phase of AAV compared with the healthy control group [12]. Nevertheless, one study conducted on 10 patients with GPA revealed that one main alteration was a 75% lower capillary density [13]. It is unlikely that these capillaroscopic alterations can be used in clinical practice due to the poor significance detected.

Tortuous capillaries: In an AAV group, tortuous capillaries were found in a comparable distribution with healthy controls [12,15]; conversely, Matsuda et al. demonstrated a higher rate of tortuous capillaries in AAV patients, which could have been due to chronic capillary injury caused by small vessels vasculitis [16].

Capillary width: There is no universal definition for capillary dilatation, but we can assume that if their width is >20 μm, they can be considered enlarged. Furthermore, capillaries with a diameter > 50 μm are regarded as megacapillaries. Enlarged capillaries were more commonly found in active vasculitis in comparison with healthy controls and non-active vasculitis [12]. In addition, cases of ectasias, even if rare, were detected only in AAV patients [15]. A recent study carried out on a cohort of 51 patients with active AAV showed that the scores of dilatated capillaries were significantly higher in the AAV group, and they were notably correlated with the disease’s activity [16]. Nevertheless, no cases of megacapillaries were detected or considered relevant by any article.

Capillary loss/avascular areas: There is no clear definition or classification of capillary loss. However, it is defined as two or more consecutive capillaries missing in the distal row [11]. Progressive capillary loss typifies the microvascular damage in systemic sclerosis; it may also be important in defining critical tissue hypoxia. As such, it can be a warning for the probable progression of avascular areas, which are associated with more extensive organ envelopment and worse prognosis of systemic sclerosis [19]. Capillary loss was significantly more recurrent in active vasculitis subjects in comparison to all the healthy controls and subjects with non-active vasculitis, suggesting a pathogenesis similar to systemic sclerosis [12,14,16]. A single study referred to capillaroscopic alterations in GPA, detecting a high proportion of avascular areas in almost 92% of patients; the study group uncovered that this alteration is a typical finding in patients with GPA [17].

Neoangiogenesis: This is seen as bizarre, ramified, or bushy capillaries [12]. It is correlated with capillary-loss-induced hypoxia and the subsequent release of vessel growth factors [10]. Several patients with active vasculitis demonstrated neoangiogenesis as a common capillary abnormality in NVC findings [12,14,16,17], with a higher rate than healthy controls and patients with non-active vasculitis. It seems to be related to the prognosis of AAV due to its potential correlation with the presence of renal damage [16].

Microhaemorrhages: Microbleeding results from disruption of the capillary wall. In almost all cases of AAV patients, nailfold haemorrhages were detected [12,13,14,15,16,17,18]. Overall, they are also common in many other connective tissue diseases and non-immune disease and in many healthy controls because they may be due to normal local trauma [17]. However, there is a recent study that linked the microhaemorrhage score to perivascular inflammatory cell infiltrations [16].

Hemosiderin deposits or stippling: Small pigment spots that can usually be found near the apical segment of the capillary, likely representing the extravasation of red blood cells from leaky capillaries, which becomes hemosiderin [12,14]. Stippling and microhaemorrhages normally overlap. Nevertheless, Keret et al. managed hemosiderin deposits as a different alteration and found that they occur only in subjects with active vasculitis [12].

Rolling: This is the disruption of flow in the capillaries. When the blood flow is slow, it becomes possible to observe packets of red blood cells moving [10]. Half of the patients with active vasculitis demonstrated slow blood flow, which resulted in a slight increase relative to the healthy controls [14,15].

Moreover, major capillaroscopy patterns can be divided into non-scleroderma and scleroderma patterns. The latter is described through a combination of enlarged capillaries, the loss of capillaries that pre-empt avascular areas, and the general disorganisation of capillaries [10]. This is commonly observed in scleroderma spectrum disorders (mixed connective tissues diseases, systemic sclerosis, and dermatomyositis) and may play a central role in the early identification and study of the outcome of these diseases. A recent publication analysed 51 patients with AAV and found that a small group (7.8%) showed drastic changes in many capillaroscopic examinations that seemed to indicate a scleroderma pattern [16].

## 4. Discussion

Vasculitis is usually characterised by the size of the blood vessel involved: large, medium, or small [21]. However, this classification is not absolute, and overlap is possible. For example, Takayasu arteritis includes mainly large arteries, yet retinal arteriovenous shunts, dilation of capillaries, and obliteration of neighbouring capillaries are characteristic [22].

The diagnosis of vasculitis can be difficult, and tissue biopsy is not always feasible. NVC may be a valuable additional instrument, permitting a direct observation at the culprit system—the vasculature [14]. However, the NVC alterations described in vasculitis are varied and not sufficiently specific to be considered clinically reliable thus far [23]. Furthermore, isolated capillary changes are frequent in healthy subjects, which increase the diagnostic challenge [24].

We resumed for the first time all the studies in the literature about the role of nailfold capillaroscopy in ANCA-associated vasculitis, following the Chapel–Hill nomenclature. A few small, uncontrolled studies have described nailfold capillary changes in vasculitis. Unfortunately, these were not comparable due to their adopting varied definitions and methods (e.g., stereomicroscope and NVC).

Among all the articles surveyed, capillaroscopic abnormalities in the AAV population were found in between 70 and 80% of patients. In numerous case series, the capillaroscopy alterations were classifiable as minor or nonspecific (e.g., increased tortuosity, microhaemorrhages, and enlarged, bushy and bizarre capillaries, and with recurrent architectonic disarrangement) [23]. Sendino et al. made use of light microscopy and described capillary alterations in 15 subjects with diverse vasculitis (i.e., ANCA-associated vasculitis, giant cell arteritis (GCA), and polyarteritis nodosa (PAN)). The authors observed isolated alterations in about 70% of patients, principally microhaemorrhages. However, these results were non-specific and did not obtain statistical significance. The conclusion was that capillaroscopy has low value in vasculitis disorders [18]. However, in other papers, some vasculitis presented major or specific variations (e.g., in granulomatosis with polyangiitis (GPA) avascular areas were reported in 92% of patients) [17]. Matsuda et al. made a stronger case for the clinical impact of NVC in AAV patients. In this study, they discovered that many capillaroscopic alterations, such as microhaemorrhage, neoangiogenesis, capillary loss, and tortuosity were significantly increased in the AAV group relative to the healthy subjects. What is interesting is the possible correlation discovered between the NVC abnormalities and the severity of organ involvement in AAV diseases. In addition, it was demonstrated that microhaemorrhages in NVC, initially evaluated at the first visit, were significantly decreased after 3 months of undergoing immunosuppressive treatment [16]. The researchers supposed that this might be due to an improvement in vascular inflammation after immunosuppressive therapy [16]. This could open the way for the use of NVC to monitor treatment.

In an interesting study, the researchers evaluated the characteristic changes (e.g., tortuosity, oedema, AND derangement) in 31 patients affected by Henoch–Schonlein purpura (HSP) [36]. In another article concerning HSP in children, the authors reported a relationship between pathology activity and NVC changes [37]. The results of this study demonstrated the normalisation of the NVC image in THE non-active phases of the disorders [37]. In their study, Movasat et al. verified an association of NVC alterations with disease stages in Behcet’s disease, where enlarged capillaries were related with younger age at pathology onset, hypertension, and superficial phlebitis [35].

The current standard device for scoring disease activity in systemic vasculitis is the Birmingham Vasculitis Activity Score (BVAS). BVAS is a certified and qualified score used to assess the significance of vasculitis manifestations in involved organs, with the capacity to distinguish between active or remission situations [25]. Accordingly, disease damage in the AAV cohort is directly linked with BVAS disease activity, such as syndrome duration, suggesting that higher disorder activity results in greater microvascular damage [15].

In general, significantly higher rates of alteration were identified in the active phase of the disease than in healthy controls and in the non-active phase [12,13,14,16,17]. For most capillaroscopy changes observed in patients in remission, those patients ended up with similar abnormality rates compared to healthy controls. Furthermore, two studies did not document any relevant NVC abnormalities in the AAV cohort compared with the healthy controls [15,20]. The study of Triggianese et al. involved a cohort of AAV patients without comorbidities, which is a rare condition in such disorders, and this could explain the results of that study. Similarly, Sullivan et al. aimed to enrol only follow-up patients with disease in remission; in this case, patients had already received therapy, and they were strictly following the treatment, where the microvascular injury was under surveillance for a long time and the vascular inflammation had probably drastically improved, justifying the results obtained. Overall, this could explain why the NVC alterations in AAV patients and healthy controls were very similar.

On the other hand, Matsuda et al. enrolled a study cohort comprising critically ill patients and mainly hospitalised ones. Arguably, the alterations detected eventually strictly correlated with AAV severity, but an observation must be made about the overlapping conditions that could happen in these critical patients: they may be more likely to exhibit nonspecific capillaroscopy alterations due to their conditions [20].

As expected, ANCA-associated vasculitis presents several and heterogeneous nailfold capillaroscopy alterations, which support microvascular damage in this group of disorders. However, as previously anticipated, most alterations detected were few and non-specific—when compared to the scleroderma pattern—except for a few studies where major and specific alterations were found [16,17]. Omitting those studies, most articles showed a significantly higher rate of active capillaroscopic abnormalities, which deserve further and larger studies.

On the other hand, some vasculitis, such as Bechet’s disease or HSP, showed major or specific changes, where connections between disorders activity, pathology duration and NVC changes were reported [35,36,37]. The real meaning of these findings is difficult to interpret, which could be due to either primary vasculitis or possible overlap with other diseases.

Regardless, the studies were non-uniform and different from each other. What could be confirmed is that nailfold capillaroscopy seems to be useful for evaluating vasculitis affecting small vessels, but further studies are needed.

## 5. Conclusions

Until now, limited data have been available regarding the clinical application of capillaroscopy in AAV. However, some studies have presented the usefulness of this tool in evaluating these disorders, especially with regard to active vasculitis. Despite this, the evidence for clinical influence is still limited, and there is a need for additional observations to support the validity of nailfold capillaroscopy. There is also an urgent need for standardisation in capillaroscopy in order to better interpret the methodology and the definitions of abnormalities. One of the main limitations of the articles reviewed is the restricted size of the AAV cohort analysed. We would like to suggest increasing the number of patients enrolled to better stratify nailfold capillaroscopy abnormalities.

## 6. Future Directions

This overview confirms the potential of NVC as a safe and non-invasive instrument for assessing the activity phase of ANCA-associated vasculitis and for evaluation of the disease during follow up. However, large amounts of work are required to confirm the clinical impact on NVC in AAC. Further studies are needed to confirm the utility of NVC in vasculitis and to demonstrate that this tool can provide important information regarding organ involvement and disease progression.

## Figures and Tables

**Figure 1 diagnostics-14-00254-f001:**
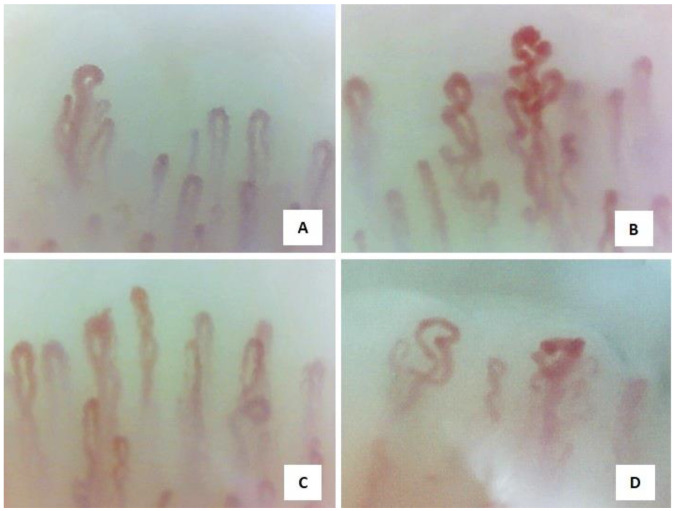
Nailfold video capillaroscopy images (×200) in four vasculitis patients: neoangiogenesis, presence of bizarre, ramified, or bushy capillaries (**A**–**D**) (operators B.R. and L. M.) [12].

**Table 1 diagnostics-14-00254-t001:** Normal capillaroscopic parameters.

**Normal Capillaroscopic Parameters**	
Capillary array, architecture, morphology	Homogeneous distribution of U-shaped capillaries, perpendicular to the nailfold
Capillary tortuosity	Usually absent
Dilatated and giant capillaries	Not present
Ramified capillaries	Not present
Neoangiogenesis	Not present
Haemorrhages, hemosiderin deposits	Usually not present, may be present after local trauma
Capillary number	Density of between 9 and 14 capillaries per mm
Avascular areas	Not present
Capillary blood flow	Dynamic, without stasis of red blood cells

**Table 2 diagnostics-14-00254-t002:** List of the selected articles reporting any possible capillaroscopy alterations in ANCA-associated vasculitis.

Selected Articles Reporting any Possible Capillaroscopy Alterations Observed				
Articles	Study Methods	Exclusion Criteria in the Vasculitis Group	Capillaroscopic Strategy and Methods	Capillaroscopy Findings
Keret et al. [12]	25 patients were examined via NVC, 17 with active vasculitis and 8 with vasculitis in remission. Among them, only 1 patient with active GPA, 3 with active EGPA, and 2 with active MPA.	Patients were excluded if they had history of onychophagia, recurrent trauma to fingers, severe metabolic vasculopathy, and CTDs.	Capillaroscopic changes were evaluated using Optilia Mediscope. NVC was performed on fingers 2, 3, 4, 5 in both hands and by only experienced examiners in all vasculitis patients.	Regarding the NVC abnormalities, they were significantly more common in patients with vasculitis in comparison with healthy controls.
Julia et al [13]	10 patients with GPA were studied using periungual capillaroscopy. All were under treatment and with active vasculitis.	Not defined.	The capillaroscope used was a stereomicroscope and perilingual capillaroscopy was performed at the level of the 3rd, 4th and 5th fingers of the right and left hands, always by the same operator.	Capillaroscopic alterations were found in 8 out of 10 patients enrolled.
Rimar et al. [14]	25 patients were evaluated by NVC, among whom there were 8 with vasculitis in remission and 7 with AAV (2 GPA, 3 EGPA, 2 MPA).	Subjects affected by peripheral artery disease and ischemic heart disease were excluded.	Capillaroscopic changes were evaluated using Optilia Mediscope.	Higher rates of non-specific capillaroscopic abnormalities were detected in the majority of study cohort.
Triggianese et al. [15]	Observational study was realised on 23 patients affected by AAV. They were analysed via NVC.	Exclusion criteria were systemic disorders such as diabetes or patients with severe renal dysfunction and other autoimmune systemic diseases (recent and previous history), neoplasia, pregnancy or lactation.	Nailfold vessel examination was performed using the NVC (Inspectis Digital Capillaroscope Light CAP-1).	82% of AAV cohort showed non-specific NVC abnormalities, which were found with similar rates compared with healthy controls.
Matsuda et al. [16]	51 patients were examined by NVC, all were newly diagnosed with AAV. NVC was performed at the first visit and was repeatedly evaluated 3 months after immunosuppressive therapy.	Patients with infectious vasculitis, drug-induced or secondary vasculitis, sarcoidosis or other CTDs and malignancy were excluded.	NVC was assessed using a Dino-lite capillaroscopy device. NVC was evaluated on fingers 2, 3, 4, 5 in both hands.	70.6% of the enrolled patients showed significant NVC changes; 4 patients even showed a scleroderma pattern.
Anders et al. [17]	They prospectively analysed of 116 patients by reviewing nailfold capillary microscopy records and made the final diagnosis of GPA for 12 patients, such that they were all newly diagnosed.	Not defined.	NCM was performed with a bifocal stereomicroscope. All fingers were examined in both hands in each patient.	Avascular areas were exclusively found in patients with GPA, in 92% of total GPA patients.
Sendino et al. [18]	Light microscopy was used to report capillary alterations in 15 patients with various vasculitis (i.e. ANCA-associated vasculitis, giant cell arteritis (GCA) and polyarteritis nodosa (PAN)).	Not defined.	Not defined.	Capillaroscopy showed changes in 11 patients, with higher rates in the active phase of the diseases.
Sullivan et al. [19]	33 patients affected by small-vessel vasculitis were recruited and they were analysed using NVC.	Patients were excluded if they had history of CTDs, type 2 diabetes, dialysis dependence, antiphospholipid syndrome, bleeding or recent hand trauma, among others.	Capillaroscopy was performed using the NVC and a minimum of 2 images per nailfold was evaluated for 8 digits (excluding thumbs) for each patient.	NVC abnormalities were detected only in 54.5% of AAV group (18/33).

## Data Availability

Data sharing not applicable.

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
