# Peer review of "Nailfold Capillaroscopy Analysis Can Add a New Perspective to Biomarker Research in Antineutrophil Cytoplasmic Antibody-Associated Vasculitis"

_diagnostics, 2024, doi:10.3390/diagnostics14030254_

Round 1

Reviewer 1 Report

Comments and Suggestions for Authors

The manuscript reports interesting data regarding the possible role of capillaroscopy in diagnostic evaluation of severity and follow up of patients with ANCA associated vasculitis.The manuscript is written in comprehensive mode . I suggest to introduce in the table 2  additional column with data regarding the number of pathological exams and /or capillaroscopy findings in enrolled patients. 

Author Response

The manuscript reports interesting data regarding the possible role of capillaroscopy in diagnostic evaluation of severity and follow up of patients with ANCA associated vasculitis.The manuscript is written in comprehensive mode . I suggest to introduce in the table 2  additional column with data regarding the number of pathological exams and /or capillaroscopy findings in enrolled patients. 

Response: Firstly, thank you for your time and effort in our article. The comments are all valuable and helpful for revising and improving our paper, as well as the important guiding significance to our researches. All added content is marked in red in the revised manuscript. In agreement with the reviewer’s comments additional column with data regarding the number of pathological exams and /or capillaroscopy findings in enrolled patients were introduced in table 2.

Reviewer 2 Report

Comments and Suggestions for Authors

An interesting review article on nailfold videocapillaroscopy in AAV.

My remarks are:

1. Nailfold video-capillaroscopy:

129 Figure 1. Nailfold videocapillaroscopy images (×200) in healthy subjects (A), “Early” (B),

“Active” (C), and “Late” (D) patterns of scleroderma microangiopathy (Operators B.R. and L. M.) [9,11].

Since the topic of the article is ANCA vasculitis, I think videocapillaroscopy images of the nail in SSc are inappropriate and confusing, so I would leave them out,

2. Discussion

 368….. more useful information regarding organ involvement and disease activity. I would leave out this part of the conclusion because not enough data was presented on the connection of capillaroscopy with the organ  involvement.

Author Response

An interesting review article on nailfold videocapillaroscopy in AAV.

Response: Thank you very much for your kind letter and for reviewers’ constructive comments concerning our article. We have revised our manuscript according to the reviewer’s comments.  The point-to-point responses have also been presented in this response letter.

My remarks are:

  1. Nailfold video-capillaroscopy: 129 Figure 1. Nailfold videocapillaroscopy images (×200) in healthy subjects (A), “Early” (B), “Active” (C), and “Late” (D) patterns of scleroderma microangiopathy (Operators B.R. and L. M.) [9,11]. Since the topic of the article is ANCA vasculitis, I think videocapillaroscopy images of the nail in SSc are inappropriate and confusing, so I would leave them out.

Response: Thank you very much for your comment. In agreement with reviewer’s suggestion Figure 1 was deleted.

  1. Discussion.368….. more useful information regarding organ involvement and disease activity. I would leave out this part of the conclusion because not enough data was presented on the connection of capillaroscopy with the organ  involvement.

Response: Thank you very much for this comment. In agreement with the reviewer's suggestions, we leave out the information